Bitcoin volatility in bull vs. bear market-insights from analyzing on-chain metrics and Twitter posts

Baroiu Alexandru Costin
http://orcid.org/0000-0002-5169-9232 Diaconita Vlad diaconita.vlad@ie.ase.ro
Oprea Simona Vasilica
Department of Economic Informatics and Cybernetics, Bucharest University of Economic Studies , Bucharest , Romania
Rosak-Szyrocka Joanna
Electronic publication date: 2023 Dec 20
Publication date: 2023
Volume: 9
Electronic Location ID: e1750
Received 2023 Apr 6; Accepted 2023 Nov 22
Copyright: © 2023 Baroiu et al.
Copyright year: 2023
Copyright holder: Baroiu et al.
License: This is an open access article distributed under the terms of the Creative Commons Attribution License, which permits unrestricted use, distribution, reproduction and adaptation in any medium and for any purpose provided that it is properly attributed. For attribution, the original author(s), title, publication source (PeerJ Computer Science) and either DOI or URL of the article must be cited.
License URL: https://creativecommons.org/licenses/by/4.0/

Keywords: On-chain analysis, Sentiment analysis, Cryptocurrency volatility, Bitcoin, LSTM, Stocks

Funding: Bucharest University of Economic Studies 323/2022 This work was supported by a grant offered by the Bucharest University of Economic Studies, project ID 323/2022, project title “Electronic voting secured by blockchain technology—applicability in university elections” and through the Ph.D. program. The funders had no role in study design, data collection and analysis, decision to publish, or preparation of the manuscript.

==============================
Cryptocurrencies have emerged as a popular investment vehicle, prompting numerous efforts to predict market trends and identify metrics that signal periods of volatility. One promising approach involves leveraging on-chain data, which is unique to cryptocurrencies. On-chain data, extracted directly from the blockchain, provides valuable information, such as the hash rate, total transactions, or the total number of addresses that hold a specified amount of cryptocurrency. Some studies have also explored the relationship between social media sentiment and Bitcoin, using data from platforms such as Twitter and Google Trends. However, the quality of Twitter sentiment analysis has been lackluster due to suboptimal extraction techniques. This research proposes a novel approach that combines a superior sentiment analysis technique with various on-chain metrics to improve predictions using a deep learning architecture based on long-short term memory (LSTM). The proposed model predicts outcomes for multiple time horizons, ranging from one day to 14 days, and outperforms the Martingale (random walk) approach by over 9%, as measured by the mean absolute percentage error metric, as well as recent results reported in literature. To the best of our knowledge, this study may be among the first to employ this combination of techniques to improve cryptocurrency market prediction.

Introduction

Blockchain has revolutionized the way people think about money, finance, governance, healthcare, technology, and many other fields (Chohan, 2017; Szabo, 2005; Satoshi Nakamoto Institute, 2004; Dai, 1998; Lamport, Shostak & Pease, 1982). The past decade has seen this transformative technology come into the limelight and create entire industries that are now worth upwards of billions of dollars (Department of Industry, Science, Energy and Resources, 2020). As such, blockchain technology has generated great interest, both in industry and academia, with price prediction of different cryptos being one of the most studied topics (Huang, Huang & Ni, 2019). The objective of this article is to contribute to the existing body of research on the prediction of cryptocurrency prices. The study seeks to enhance understanding of the underlying mechanisms that influence fluctuations in crypto assets’ prices. Additionally, the article seeks to elucidate the similarities and differences between the crypto market and traditional financial markets. Furthermore, the investigation intends to develop high-performance prediction models that can facilitate the generation of wealth by financial experts. The findings can also assist regulators in comprehending the mechanisms that drive crypto assets, thereby enabling the drafting of better regulation.

Since its inception, Bitcoin (Nakamoto, 2008) has been viewed by some as a lucrative investment opportunity for early adopters willing to take on high risks. Although its volatility has deterred some potential investors, others recognize its potential for significant upside gains. Bitcoin has been widely touted by its supporters as a potential alternative to gold, believed to be a hedge against inflation, and considered to have played a significant role in the financial advancements of the 21st century. While other applications remain important, the investment potential of cryptocurrencies is their most important quality at the time of writing and the main trait that is studied in this article.

Price analysis and prediction of cryptocurrencies are an active research field, in academia and in industry. The importance of price predictors was highlighted by Kliestik et al. (2020), who showed that predictors can be used to assess financial health and eliminate potential risks. Morales, Gray & Rajmil (2022) and Mohan et al. (2023) aim to address FinTech specific challenges by leveraging advanced techniques, such as machine learning and optimization algorithms, to improve accuracy and efficiency in areas like financial crisis prediction and risk assessment. Additionally, their findings underscore the importance of a robust regulatory framework to ensure effective FinTech governance and sustainable growth in the face of rapid industry development. Cryptocurrencies are also exposed to risks, as presented by Schinckus, Nguyen & Chong (2021). The authors found that there is a significant relationship between pseudonymous currencies like Bitcoin or Ethereum (Buterin, 2015) and anonymous cryptocurrencies. DASH is identified as a key cryptocurrency that could be involved in the dynamics of Bitcoin and Ethereum. Prediction systems can help solve these issues by offering a better understanding of the evolution of different phenomena and their inner workings. To this end, much research has been done on the task of predicting asset prices.

The work presented in this article builds on the research done by Shen, Urquhart & Wang (2019), which has shown that Twitter volume is a superior predictor of Bitcoin price compared to Google Trends, and the work of Sattarov et al. (2020), who analyzed the correlation between Bitcoin and Twitter sentiment and built a random forest (RF) predictor that achieved a reported accuracy of 63%. More recently, Critien, Gatt & Ellul (2022) integrated both Twitter sentiment and volume to predict the direction of the price of Bitcoin. An interesting note to make is that both Sattarov et al. (2020) and Critien, Gatt & Ellul (2022) used Valence Aware Dictionary and Sentiment Reasoner (VADER) (Hutto & Gilbert, 2014) to extract sentiment from Twitter data, which is a lexicon and rule-based analyzer. Besides Twitter data, on-chain metrics have also been employed in cryptocurrency price prediction systems. Jagannath et al. (2021) used 26 on-chain metrics to train a long short-term memory (Hochreiter & Schmidhuber, 1997) architecture for Ethereum price prediction.

This article advances recent research by employing a more advanced sentiment extraction method (Lecun et al., 1998). To our knowledge, it is the first to concurrently use a larger set of on-chain metrics and sentiment data from Twitter for optimal prediction results. We seek to improve the work done by Jagannath et al. (2021) by adding more than double the on-chain metrics to the analysis, 54 compared to 26, and by expanding the study of the relationship between these metrics and Bitcoin. To assess the efficiency of our approach, the following research questions are postulated:

RQ 1. What are the on-chain metrics that exert the most influence on the Bitcoin price and how many of these metrics can signal a 5% variation of price?

RQ 2: Is the influence of Twitter sentiment and on-chain metrics on Bitcoin price uniform across different market conditions (e.g., bull markets, bear markets, stable periods)?

RQ 3. To what extent does Twitter sentiment influence Bitcoin price and what is the lag between sentiment and price?

RQ 4: How does the combination of on-chain metrics and Twitter sentiment improve the accuracy of Bitcoin price prediction models?

The article is structured as follows: We begin by discussing recent and relevant related works and propose our research questions. Next, we outline the methodology employed in the study. This is followed by a presentation of the data sets used, as well as the necessary transformations. We then detail the experimental setup, which leads into a comprehensive analysis of the results and associated discussion. To conclude, we explore the theoretical contributions, managerial implications, limitations, and potential avenues for future research.

Related works

Technical analysis (TA) is a methodology that uses historical data, like stock price and volume, to anticipate future price movements (Lo, Mamaysky & Wang, 2000). Svogun & Bazán-Palomino (2022), by using TA, found that bubble periods increased the likelihood of Ethereum, Ripple, and Litecoin beating buy-and-hold, but less so for Bitcoin and Bitcoin Cash. Additionally, transaction costs decreased this likelihood for Ripple and Litecoin, but increased it for Bitcoin and Ethereum. The findings suggest that transaction costs exert a stronger influence on profitability in shorter timeframes, like 1 min, but this influence diminishes in longer timeframes, such as a day. Therefore, while TA has been effective in predicting Bitcoin prices in the short term, further research is necessary to better understand its effectiveness in longer time spans, such as 1 day or several days.

In addition to TA, there is another type of analysis that is exclusive to blockchain currencies, on-chain analysis. This technique allows the use of blockchain data, such as hash rate, block height, or transaction volume, to determine future prices. On-chain metrics have proven to offer valuable information about Bitcoin and recent studies have validated their importance. Researchers have approached on-chain analysis as a method to improve prediction performance. One early study integrated blockchain information into a Bayesian neural network (BNN) (Jang & Lee, 2018). Some of the on-chain data used in the study was average block size, transactions per block, hash rate, difficulty, miners’ revenue, and cost % of a transaction. The authors also included in their study macroeconomic indicators, like the S&P 500, and global currency exchange rates with the USD. Another study used on-chain data to characterize the Bitcoin and Ethereum markets and make price predictions (Saad & Mohaisen, 2018). The authors used public APIs to collect the blockchain data of both currencies. For each asset, 10 on-chain features were extracted, such as hash rate, miners’ revenue, difficulty, or transaction fee. For prediction, the authors employed three approaches: regression, LSTM and conjugate gradient. Jay et al. (2020) proposed a stochastic model that showed improvements over deterministic models. By introducing randomness into the neural network, the authors were able to achieve better results. The training data for the model were composed of 23 features, extracted from the blockchain and from Twitter and Google Trends. The authors used Twitter and Google search volume to quantify market sentiment.

A similar approach was implemented by Wang, Shen & Li (2022). The article investigates the role of investor attention in affecting Bitcoin prices and returns, leveraging LSTM for the analysis. It introduces an aggregate proxy for investor attention that combines both direct and indirect indicators. The empirical findings reveal that incorporating attention variables can significantly improve the predictive accuracy of LSTMs in forecasting Bitcoin returns. Among these attention variables, direct proxies like Google Trends and Tweets appear to offer more valuable information for enhancing the model’s forecasting capabilities.

A recent study applied an on-chain approach to predict Ethereum prices (Jagannath et al., 2021). The authors seem to be the first to use self-adaptive algorithms in deep learning in conjunction with on-chain data to predict the price evolution of cryptocurrencies. The research used 26 on-chain metrics and developed an LSTM model using three optimization techniques: L-SHADE optimization algorithm (Tanabe & Fukunaga, 2014), jSO optimization algorithm (Brest et al., 2006) and multi-population-based ensemble of mutation strategies (Wu et al., 2016). The authors concluded that on-chain metrics are a good supplementary tool to existing deep learning techniques when it comes to cryptocurrency price prediction. Most recently, Chen (2023) developed a Bitcoin price prediction system using an RF model and 47 metrics, divided into eight categories: Bitcoin price variables, technical features of Bitcoin, other cryptocurrencies, commodities, market index, foreign exchange, public attention, and dummy variables of the week. The model was trained on two periods, one ranging from April 2015 to October 2018 and the other from October 2018 to April 2022. The author showed that the RF model achieved results superior to those of an LSTM model. Due to the recency of the article and the improved performance of the presented approach, the results reported in this article will be used to benchmark the Proposed Approach in our article.

Twitter sentiment has also been previously used in literature, with varying degrees of success. Mittal et al. (2019) predicted short-term Bitcoin price fluctuation by implementing web search and social media data. The authors found that while Google Trends and Twitter volume were correlated with short term price volatility, Twitter sentiment did not seem to have much of an effect. The authors used VADER to extract the sentiment from the tweets. Twitter volume was again shown in another study to be effective in predicting Bitcoin price, to the detriment of Google Trends (Shen, Urquhart & Wang, 2019). The authors demonstrated that volatility and the next day trading volume were driven by the volume of tweets on Bitcoin.

One study found that linguistic features extracted from tweets can be used to improve the prediction of sudden changes in cryptocurrency prices (Sekioka, Hatano & Nishiyama, 2023). The study used Sentence-BERT to create features from tweets and trained a light-gradient boosting machine (LightGBM) model to predict price changes. The results showed that the linguistic features were able to improve the prediction accuracy, suggesting that social media data can be a valuable source of information for predicting cryptocurrency trends.

Researchers also studied the effect of COVID-19 on social media sentiment and Bitcoin price volatility. Bejaoui et al. (2021) focused on understanding the intricate relationships between Bitcoin prices, social media metrics, and the COVID-19 pandemic. Utilizing various statistical models, including fractional autoregressive vector models and fractional error correction models, the article analyzes daily data from December 31, 2019, to October 30, 2020. The findings revealed both short-term and long-term connections between Bitcoin prices, social media activity (measured through Tweets and Google Trends), and the severity of the COVID-19 crisis. The article noted that the pandemic appears to have encouraged investment in digital currencies like Bitcoin.

Another article that tackled the COVID-19 pandemic (Bouteska, Mefteh-Wali & Dang, 2022) investigated the influence of investor sentiment on Bitcoin returns, utilizing a sentiment index crafted through computational text analysis and principal component analysis (PCA). The authors applied vector autoregressive analysis among other methodologies to explore the relationship between this sentiment index and Bitcoin returns. Their findings suggest that investor sentiment is a powerful predictor of short-term cryptocurrency market returns. Additionally, the research confirms that during the COVID-19 pandemic, investors’ sentiments had a significant impact on Bitcoin returns. The authors state that the sentiment index can enable investors to generate excess returns if used for predictive purposes.

Besides Twitter or Google Trends, other sources of sentiment have been used in research. One article examined the correlation between investor sentiment expressed on StockTwits, a social media platform for investors, and intraday Bitcoin returns (Guegan & Renault, 2021). Through analysis of around one million messages, the study found a statistically significant relationship between investor sentiment and Bitcoin returns for frequencies up to 15 min. Beyond this timeframe, the relationship becomes insignificant. The study also noted that the influence of sentiment on returns is particularly prominent around the period of the Bitcoin bubble. Despite these findings, the effect’s magnitude is deemed too small to enable traders to make economic profits based solely on social media information.

Another article delved into the impact of media coverage on Bitcoin market dynamics, particularly focusing on its role in bubble formation (Li et al., 2022). Three key insights are gleaned from the research. First, any media coverage, irrespective of its tone, boosts Bitcoin returns the next day during a bubble period but not otherwise. Second, Bitcoin returns can themselves forecast the extent of subsequent media coverage in both bubble and non-bubble periods. Lastly, the relationship between media coverage and Bitcoin’s next-day trading volume is insignificant during bubble periods but negatively correlated in non-bubble times.

A study by Gherghina & Simionescu (2023) investigated the asymmetric effect of COVID-19 pandemic news on the cryptocurrency market. Using daily data from January 2020 to September 2021, the authors found that both adverse and optimistic news had the same effect on Bitcoin returns, indicating that fear of missing out behavior does not prevail. The analysis also showed that both positive and negative shocks in pandemic indices promote Bitcoin’s daily changes, suggesting that Bitcoin is resistant to the pandemic crisis and may serve as a hedge during market turmoil. The empirical results indicate that pandemic news could significantly influence Bitcoin’s price.

Another study examined the impact of investor attention due to the COVID-19 pandemic, Twitter-based sentiment towards uncertainty and public sentiment on the performance of cryptocurrencies (Bashir & Kumar, 2023). The authors used simple linear regression, quantile regression, the exponential generalized autoregressive conditional heteroskedasticity (EGARCH) model, and sentiment analysis to examine this phenomenon. They found that investor attention and Twitter uncertainty have a negative effect on cryptocurrency returns. The quantile regression results indicated that the effect of investor attention and Twitter economic uncertainty on cryptocurrency returns is heterogeneous, with a higher effect in the lower quantiles. The findings suggest that cryptocurrencies failed to act as a haven during the COVID-19 pandemic. These findings are in contradiction with the ones reported by Gherghina & Simionescu (2023).

Despite the various approaches used in the literature, some gaps can be observed from the analysis of the presented research. Specifically, most approaches used a smaller number of on-chain metrics (Jang & Lee, 2018; Saad & Mohaisen, 2018), with only recent works increasing the number of analyzed metrics to over 20 (Jay et al., 2020; Jagannath et al., 2021). Additionally, while on-chain metrics have been used in tandem with other data, only one study (Jay et al., 2020), introduced data from Twitter and it was volume data, not sentiment data, which has proven to improve prediction accuracy. However, the present study aims to address these gaps by introducing more on-chain metrics to improve prediction quality and increase understanding of Bitcoin price evolution, using Twitter sentiment alongside volume, and using a superior sentiment extraction technique compared to previous studies. Furthermore, larger time windows will be devised for price prediction, ranging from one to seven days, unlike the more limited time window used in previous studies. This measure was introduced to tackle the time sensitivity reported in research. Many articles find significant relationships in the short term but don’t explore why these relationships might fade over longer periods or if a model could be devised that can handle long term predictions. The work presented in this article aims to predict on a larger time horizon, where the task becomes more challenging.

Materials and Methods

To answer the research questions, we conduct a quantitative study. To this end, on-chain metrics, together with Twitter sentiment, are being used to determine the best approach for Bitcoin price prediction. This section will thoroughly present the concepts used in the current research and how they will blend to significantly improve the results.

On-chain analysis

The present study employs on-chain metrics to enhance the accuracy of a deep learning model for predicting cryptocurrency prices. To our knowledge, this study employs the largest number of on-chain metrics, namely 54, compared to previous studies. All on-chain data were collected from the public Bitcoin blockchain, utilizing application programming interfaces from online resources such as Glassnode (2022). To provide a comprehensive understanding of on-chain data and its potential for predicting cryptocurrency prices, a subset of 14 out of the 54 selected metrics is described in Table 1 along with their corresponding online resource descriptions.

Table 1 On-chain data metrics overview.

No. Crt.	On-chain metric	Description	
1	Active addresses	Number of unique addresses active as sender or receiver.	
2	Addresses with non-zero balance	Number of unique addresses holding a positive number of coins.	
3	New addresses	Number of unique addresses appearing for the first time in a coin transaction.	
4	Miner revenue fees	Percentage of miner revenue derived from fees.	
5	Coin days destroyed	Measure of the age of coins spent in a transaction.	
6	MVRV ratio	Ratio between market cap and realised cap.	
7	NVT ratio	Ratio between market cap and on-chain transaction volume.	
8	SOPR	Ratio of the realised value to the creation value of a wasted output.	
9	NUPL	Difference between relative unrealized profit and relative unrealized loss.	
10	Difficulty	Estimated number of hashes required to mine a block.	
11	Hash rate	Estimated number of hashes per second produced by miners.	
12	Transaction count	Total number of successful transactions.	
13	UTXO total	Total number of unspent transaction outputs.	
14	Market cap	Product of current supply and USD price	

Sentiment analysis

A machine learning approach is used to determine the sentiment of extracted tweets that builds upon previous research methodologies. Some researchers (Jagannath et al., 2021) dismissed sentiment analysis and used tweet volume for sentiment substitute, while others (Oprea et al., 2021) used basic sentiment analysis techniques. For this research, logistic regression (LR) is the preferred model for sentiment analysis, which predicts binary data such as positive or negative sentiment. LR is a simpler and less time-consuming technique, still widely used for sentiment analysis tasks (Tyagi & Sharma, 2018).

The Sentiment140 dataset, a balanced dataset of 1.6 million tweets, is used to train the LR model. The tweets are preprocessed by lowercasing the text, eliminating special characters, numbers, emojis, stemming, and lemmatizing the words. The dataset is split into training and testing sets with an 85/15 split. Figure 1 highlights the performance of the LR model.

Figure 1 Confusion matrix and area under curve (AUC) for the LR model.

The LR model achieved an 83% AUC score, signifying a strong performance for a complex task. Additionally, the model demonstrated balanced results, implying a robust F1 score. For a more comprehensive evaluation, we compared the LR model’s performance with that of VADER, a sentiment analysis tool often cited in prior research. The results for VADER are detailed in Fig. 2.

Figure 2 Confusion matrix and AUC for VADER.

Upon comparison, VADER’s performance fell short, with an AUC score of only 67–16% lower than the LR model. VADER appeared to particularly struggle with the over-identification of positive sentiments, known as False Positives. Based on these findings, we have determined that the LR model outperforms VADER and will be our chosen method for extracting sentiment from Bitcoin-related tweets.

Long-short term memory model

A long-short term memory (LSTM) network, which is a type of recurrent neural network (RNN), has been chosen for developing a predictive model. LSTM models are commonly used for time series prediction tasks and can capture long-term dependencies in data for accurate predictions. Previous studies have applied LSTM networks to predict asset prices (Moghar & Hamiche, 2020), cryptocurrencies (Jagannath et al., 2021; Jay et al., 2020; Lahmiri & Bekiros, 2019), and the number of COVID-19 cases (Chimmula & Zhang, 2020).

LSTM networks consist of a memory cell and three gates (input, output, forget) that channel information through the layers. These gates use a sigmoid activation function to process the information and ensure that only positive values are passed to subsequent gates. The forget gate decides what information will be erased from the cell’s state, while the input gate creates a candidate potential vector using an activation layer. The old state of the cell is updated, and the resulting hidden state is multiplied with the filtered output to obtain the hidden state for the next cell. The LSTM model operates on different time steps within each block and passes outputs to the next block until the final LSTM block generates the sequential output.

The LSTM network assumes a mini-batch size of n, the number of inputs as d, and h hidden units. Gates are defined as It ∈ Rn×h (input gate), Ft ∈ Rn×h (forget gate), and Ot ∈ Rn×h (output gate) and are calculated using Eqs. (1)–(3).

Thus, the input is Xt ∈ Rn×d and the hidden state of the last step is Ht−1 ∈ Rn×h. Accordingly, the gates are defined as follows: the input gate is It ∈ Rn×h, the forget gate is Ft ∈ Rn×h, and the output gate is Ot ∈ Rn×h. They are calculated as follows:

(1) It=σ(XtWxi+Ht−1Whi+bi)

(2) Ft=σ(XtWxf+Ht−1Whf+bf)

(3) Ot=σ(XtWxo+Ht−1Who+bo)

where Wxi,Wxf,Wxo ∈ Rd×h and Whi,Whf,Who ∈ Rh×h are the weight parameters and bi, bf, bo ∈ Ri×h are bias parameters. The candidate potential vector Ct is computed using Eq. (4), and the old state of the cell is updated using Eq. (5).

(4) C~t=tanh(XtWxc+Ht−1Whc+bc)

where Wxc ∈ Rd×h and Whc ∈ Rh×h are the weight parameters and bc∈ Ri×h is the bias parameter. Then, the old state of cell Ct−1 is updated as follows:

(5) Ct=Ft×Ct−1+It×C~t

Finally, the hidden state ht is obtained by multiplying the state of the scaled cell by the filtered output, as shown in Eq. (6).

(6) ht=Ot×tanh(Ct)

Proposed approach

The proposed approach involves several steps, including the collection of Twitter data and extraction of sentiment, the collection and pre-processing of on-chain data, and analysis of the relationship between the data and Bitcoin price using Pearson correlation and time series plots. The LSTM model is trained with different time steps, moving average windows, and hyperparameters, and evaluated using root mean squared error (RMSE), mean absolute error (MAE), and mean absolute percentage error (MAPE) metrics. The Martingale model serves as a baseline for comparison and is trained only on price data. The equations for RMSE, MAE and MAPE are depicted below:

(7) RMSE=1n∑k=1n(yk−y^k)2

(8) MAE=1n∑k=1n|yk−y^k|

(9) MAPE=100%n∑k=1nyk−y^kyk

where n is the number of observations and yk and y^k are the real and predicted values of kth point. The proposed flow is shown in Fig. 3. For a better representation of the PA and for better replication purposes, the steps of the PA are presented below. 1. Data collection. a. Cryptocurrency price (source: CoinMarketCap).

b. On-chain metrics (source: Glassnode).

c. Social media sentiment (source: Twitter).

2. Data preprocessing. d. Data cleaning (treat NA Values, for text data remove stop words, stemming, lemming).

e. Data normalization (apply z-normalization).

f. Time series decomposition (ensure stationarity with Augmented Dickey-Fuller–differencing is applied).

g. Sentiment analysis (train a logistic regression model on a public data set available on Kaggle–Sentiment140 in this example–and extract sentiment from Twitter Data).

3. Exploratory data analysis. h. Univariate analysis (descriptive statistics for each metric).

i. Multivariate analysis (explore the relationship between metrics–Compute Pearson Correlation for all the selected variables).

j. Time series plots (analyze the price evolution of cryptocurrencies).

k. Heatmaps (plot the correlation matrix in an intuitive manner to facilitate analysis).

4. Model building–neural network with two LSTM layers and softmax activation function in this example. l. Time step selection (iteratively train models for different time steps 1–14 days in this study).

m. Moving average selection (iteratively train models for different moving average ranges).

n. Feature selection with correlation threshold (establish correlation thresholds and iteratively select only those variables over the threshold value).

o. Data split (split the data in train-test-validation data sets in 70-15-15 split).

p. Hyperparameter tuning (iteratively train the model with different layer size, learning rate and dropout values).

q. Performance evaluation (run the model on the test set and benchmark the results to identify the best performance).

Figure 3 The proposed research flow.

The codebase used for this study can be found in the following Zenodo repository, which follows the logical flow presented above: https://doi.org/10.5281/zenodo.7791503.

Data

As previously stated, we used multiple datasets in this investigation. We constructed the Twitter dataset using the snscrape Python package, which serves as a scraper for social networking services. We collected the tweets from July 1, 2021, until June 30, 2022. There are 1,000 tweets collected per day, for a total of 365,000. Two restrictions are imposed on the scraper, i.e., the tweets had to be in English and they must not have been retweets. By excluding retweets, the aim is to maintain the independence of data points in the generated sample. Retweets often serve as duplicates of original messages and including them could introduce bias or artificially inflate the significance of sentiments or phrases in the analysis. The first entries of the resulting database are presented in Table 2.

Table 2 Twitter sentiment data example.

User	Date created	No of likes	Source of tweet	Tweet	
Coincolumnist	2021-07-01 23:59:58+00:00	0	Blog2Social APP	Mercado Bitcoin, the largest crypto exchange in Brazil, has received a $200 million investment from the Japanese multinational conglomerate holding company, Softbank Group Corp, in a recently-concluded Series B funding. #AANews #Banks #bitcoin #Brazil https://t.co/OkMTkZyAcm	
CryptoWatchBot	2021-07-01 23:59:53+00:00	2	Crypto watch bot	#Investing 100.0% in this optimal #crypto portfolio and −0.0% #cash in the past 24 h would have given you a −3.6% return rather than #Bitcoin’s −4.0%, for the same level of #risk https://t.co/aLNaqNR8yt	
KevinKrypto	2021-07-01 23:59:53+00:00	0	Twitter web app	@michael_saylor Bitcoin Mining Council… well a very impressive name indeed. One of the “cleanest industries in the world”… wow that is impressive too… they surveyed the network did they… interesting. Well case closed then. BTC is as green as grass	
FurryDramaBot	2021-07-01 23:59:49+00:00	5	Cheap bots, done quick!	Krinkels tried to scam a dog via pigeon post, saying: ‘You will die of the bubonic plague unless you send Bitcoin to this wallet’ https://t.co/JHO4Wku4Fm	
PhoneHustles	2021-07-01 23:59:45+00:00	0	Twitter for iPhone	The Easiest Way To Invest In Bitcoin GET $10 of free #BITCOIN #cryptocurrency #invest #passiveincome #makeyourmoneymakemoney #phonehustles	

Twitter data are enriched with the volume of tweets per day on the topic of Bitcoin, extracted from BitInfoCharts (bitinfocharts.com). After the Twitter data has been collected, the sentiment is extracted from the tweets. There are multiple pre-processing steps applied to the text. The selected model is LR, which outperformed VADER for this specific use case. The resulting data set is numeric, with binary data for sentiment (1 for positive or 0 for negative) and continuous data for volume. After the Twitter data is computed, the on-chain data are extracted and the working data set is created, joining the Twitter data and the on-chain data. On the working data, there are several preprocessing steps to ensure that the time series data is appropriate for analysis. First, all the data sets are normalized by applying z-normalization as in Eq. (7).

(10) zscore=x−μσ

where x is the original value, μ is the mean and σ is the standard deviation. The study focused on the shift from a bull market to a bear market during the period from July 1, 2021, to June 30, 2022. The complete set of data used for this study can be found in the following Zenodo repository: https://doi.org/10.5281/zenodo.7791503.

The bull market period covered the first 131 days from July 1, 2021, to the all-time high (ATH) date, while the bear market period covered the last 234 days starting from the ATH date of November 8, 2021. An inversion period of 30 days before and after the ATH was selected to better understand the signal for the bull-to-bear transition. The selection of this period was deliberate, aimed at studying the distinct mechanisms that drove both bull and bear market phases, as well as the factors contributing to the transition between them. Understanding these shifts is critical for investors, policy-makers, and researchers alike as it helps to navigate the volatile cryptocurrency market more effectively. The aim of this article is to develop a robust predictive model capable of navigating these market shifts, thereby providing a more comprehensive understanding of Bitcoin’s price volatility across different market conditions.

The correlation matrix is used to identify on-chain metrics that have the most influence on Bitcoin price, answering RQ1 (the first part), and to analyze the three scenarios, answering RQ2. Only metrics with a Pearson correlation coefficient greater than 0.8 are presented in the correlation matrix for each scenario. This cutoff was selected based on common guidelines in statistical literature, which generally regard a correlation above 0.8 as very strong. This high threshold was used to ensure that only the most highly correlated variables were included in the initial model, to reduce multicollinearity and improve model performance. The bull market matrix is shown in Fig. 4.

Figure 4 Bull market correlation matrix.

A high positive correlation exists between different metrics, some expected as they are derived from the price or each other, while others are block or mining metrics. These metrics indicate increased network interest and could be strong bull market indicators. Wallets with more than 0.1 Bitcoin show a high correlation with other metrics, suggesting an increasing number of new investors. Bitcoin Fund Holdings is the only metric presenting a negative correlation with all other metrics and could anticipate a drop as the market rushes toward the ATH, which could indicate an impending drop or a successful gamble.

In the bear market correlation matrix presented in Fig. 5, there is a tonal shift with many negative correlations observed. Bitcoin Fund Holdings, Hash Rate, and Supply Last Active 5+ years ago are absent, possibly indicating a cool-off in Fund activity and volatile market state with investors reacting by freezing their holdings or buying more Bitcoin. The negative correlation between small holding wallets and price suggests that new or inexperienced investors are trying to enter the market at a discount. UTXO points towards high market volatility with more transactions executed and sell orders put out. As such, RQ2 is answered as it can be noticed that on-chain metrics and Twitter sentiment exert different influences in different market conditions. This finding is aligned with previous research.

Figure 5 Bear market correlation matrix.

The inversion period correlation matrix in Fig. 6 reveals interesting findings, with a decrease in correlation between MVRV and NUPL and Market Cap and Price. MVRV is almost perfectly correlated with the two metrics in both bull and bear markets, but the inversion period shows a decrease in correlation. The absence of all the other metrics could be another sign of shifting behavior. Future studies could investigate present actions during a bear-to-bull market shift to determine if they align with this study’s findings.

Figure 6 Inversion correlation matrix.

To identify what metrics could signal a 5% price variance of Bitcoin and answer the second part of RQ1, the correlation matrix is computed. Only days that precede price changes of more than 5% are selected. The same filters as before are applied, and only metrics that present a correlation coefficient greater than 0.8 are selected. The resulting matrix, presented in Fig. 7, closely resembles the correlation matrix of the inversion period. This similarity points toward a sign of inversion when the price swings greater than 5%, it becomes more frequent. As in the case of market inversion, the same metrics are present: Market Cap to Thermocap Ratio, MVRV, NUPL and Market Cap. High volatility of these metrics’ points toward high volatility of the price the following day.

Figure 7 The 5% Correlation matrix.

Next, to better understand how price relates to various other metrics over time, particularly Twitter data, we will present the Pearson correlation coefficients for each metric at different time lags. This analysis aims to address Research Question 3 (RQ3). The relevant data for this can be found in Table 3. First, it can be noted that Twitter data have little influence on the present price of Bitcoin. However, as the price is lagged further into the future, Twitter data becomes more significant. The correlation between the volume of tweets and price increases the greater the time difference between the two. From lag 7 onwards, the coefficient becomes statistically significant, and the value reaches a maximum of 0.49. The two metrics are negatively correlated. This could indicate how long it takes for the price to assimilate discussions and activity on Twitter. For Twitter sentiment, there seems to be no correlation present, for any lag of price, answering RQ3. This finding is consistent with previous research, and Twitter could not be a reliable source of Bitcoin sentiment. Future studies could look for other, more niche sources of data, like Internet forums. However, the problem of sentiment extraction remains, and future studies could further improve the sentiment analysis approach shown in this research. In Natural Language Processing (NLP) research, transformer-based architectures have proven to net excellent results and could prove beneficial for this task.

Table 3 Pearson correlation coefficient with price (statistical significance, p-value).

Metric	Present	1	7	14	21	30	45	
Price	1 (1)	0.989 (0)	0.924 (0)	0.836 (0)	0.73 (0)	0.596 (0)	0.372 (0)	
Market cap	1 (0)	0.989 (0)	0.922 (0)	0.831 (0)	0.722 (0)	0.585 (0)	0.357 (0)	
MVRV	0.946 (0)	0.941 (0)	0.914 (0)	0.876 (0)	0.826 (0)	0.755 (0)	0.622 (0)	
Market cap to thermocap ratio	0.929 (0)	0.925 (0)	0.902 (0)	0.87 (0)	0.83 (0)	0.767 (0)	0.652 (0)	
NUPL	0.899 (0)	0.896 (0)	0.873 (0)	0.857 (0)	0.83 (0)	0.778 (0)	0.648 (0)	
Inter-exchange transfers	0.655 (0)	0.649 (0)	0.605 (0)	0.557 (0)	0.533 (0)	0.454 (0)	0.306 (0)	
Bitcoin fund holdings	0.57 (0)	0.569 (0)	0.564 (0)	0.543 (0)	0.471 (0)	0.737 (0)	0.767 (0)	
SOPR	0.514 (0)	0.504 (0)	0.507 (0)	0.507 (0)	0.444 (0)	0.364 (0)	0.248 (0)	
Over 100	0.508 (0)	0.501 (0)	0.455 (0)	0.401 (0)	0.336 (0)	0.25 (0)	0.101 (0.071)	
3iq holdings	0.501 (0)	0.485 (0)	0.344 (0)	0.1 (0.061)	−0.107 (0.046)	−0.001 (0.983)	−0.225 (0)	
New addresses	0.383 (0)	0.381 (0)	0.315 (0)	0.252 (0)	0.185 (0.001)	0.085 (0.118)	−0.112 (0.044)	
Exchange balance	0.355 (0)	0.352 (0)	0.337 (0)	0.319 (0)	0.349 (0)	0.355 (0)	0.412 (0)	
Transaction count	0.294 (0)	0.291 (0)	0.237 (0)	0.18 (0.001)	0.126 (0.019)	0.052 (0.343)	−0.083 (0.141)	
Exchange net position change	0.281 (0)	0.269 (0)	0.181 (0.001)	0.034 (0.528)	−0.031 (0.563)	−0.052 (0.346)	0.043 (0.443)	
Receiving addresses	0.271 (0)	0.268 (0)	0.191 (0)	0.114 (0.033)	0.043 (0.424)	−0.053 (0.335)	−0.232 (0)	
Active addresses	0.239 (0)	0.232 (0)	0.169 (0.001)	0.093 (0.083)	0.055 (0.306)	−0.029 (0.599)	−0.168 (0.003)	
Inflation rate	0.224 (0)	0.23 (0)	0.194 (0)	0.221 (0)	0.186 (0.001)	0.167 (0.002)	0.087 (0.122)	
Over 10	0.21 (0)	0.216 (0)	0.24 (0)	0.268 (0)	0.235 (0)	0.141 (0.01)	0.059 (0.291)	
Blocks mined	0.201 (0)	0.207 (0)	0.168 (0.001)	0.192 (0)	0.153 (0.004)	0.133 (0.015)	0.05 (0.374)	
Issuance	0.201 (0)	0.207 (0)	0.168 (0.001)	0.192 (0)	0.153 (0.004)	0.133 (0.015)	0.05 (0.374)	
Sending addresses	0.183 (0)	0.18 (0.001)	0.144 (0.006)	0.097 (0.069)	0.083 (0.125)	0.03 (0.578)	−0.05 (0.377)	
Free ratio multiple	0.14 (0.008)	0.14 (0.008)	0.103 (0.052)	0.069 (0.197)	−0.034 (0.528)	−0.09 (0.099)	−0.188 (0.001)	
NVT	0.131 (0.012)	0.134 (0.011)	0.118 (0.026)	0.117 (0.029)	0.107 (0.047)	0.161 (0.003)	0.174 (0.002)	
Exchange withdrawals	0.108 (0.039)	0.109 (0.038)	0.074 (0.16)	0.082 (0.127)	0.146 (0.007)	0.099 (0.071)	0.003 (0.959)	
Transaction size	0.105 (0.045)	0.099 (0.058)	0.076 (0.149)	0.018 (0.739)	0.006 (0.918)	−0.063 (0.248)	−0.133 (0.017)	
Suply last active 2+ years ago	0.098 (0.062)	0.103 (0.05)	0.15 (0.004)	0.251 (0)	0.383 (0)	0.544 (0)	0.587 (0)	
Tweets sentiment	−0.019 (0.717)	−0.015 (0.775)	0.02 (0.708)	0 (0.994)	0.004 (0.939)	−0.007 (0.896)	0.04 (0.472)	
ASOL	−0.021 (0.691)	−0.03 (0.572)	0.225 (0)	0.206 (0)	0.251 (0)	0.159 (0.004)	0.181 (0.001)	
Coin days destroyed	−0.04 (0.451)	−0.045 (0.396)	−0.019 (0.714)	0.019 (0.723)	0.024 (0.66)	0.02 (0.722)	−0.049 (0.386)	
Transfer volume total	−0.042 (0.422)	−0.042 (0.424)	−0.101 (0.056)	−0.09 (0.093)	0.042 (0.435)	−0.051 (0.355)	−0.2 (0)	
Block size mean	−0.042 (0.421)	−0.051 (0.329)	−0.047 (0.37)	−0.113 (0.035)	−0.099 (0.068)	−0.143 (0.009)	−0.148 (0.008)	
Tweets volume	−0.065 (0.218)	−0.073 (0.162)	−0.122 (0.02)	−0.14 (0.009)	−0.179 (0.001)	−0.273 (0)	−0.492 (0)	
MSOL	−0.098 (0.061)	−0.1 (0.056)	0.05 (0.341)	0.016 (0.769)	0.059 (0.272)	0.015 (0.79)	0.012 (0.828)	
Transfer volume mean	−0.099 (0.059)	−0.098 (0.063)	−0.148 (0.005)	−0.121 (0.024)	0.024 (0.655)	−0.057 (0.295)	−0.187 (0.001)	
Inter-exchange volume	−0.119 (0.023)	−0.116 (0.027)	−0.113 (0.032)	−0.056 (0.294)	0.053 (0.324)	0.023 (0.677)	−0.089 (0.113)	
Fees total	−0.179 (0.001)	−0.181 (0.001)	−0.185 (0)	−0.167 (0.002)	−0.058 (0.285)	0.021 (0.702)	0.114 (0.041)	
Supply last active 5+ years ago	−0.182 (0)	−0.185 (0)	−0.189 (0)	−0.206 (0)	−0.26 (0)	−0.327 (0)	−0.365 (0)	
Miner revenue fees	−0.194 (0)	−0.195 (0)	−0.195 (0)	−0.186 (0)	−0.089 (0.099)	−0.009 (0.875)	0.092 (0.102)	
Block interval mean	−0.211 (0)	−0.216 (0)	−0.181 (0.001)	−0.199 (0)	−0.156 (0.004)	−0.122 (0.026)	−0.031 (0.582)	
Exchange inflow volume	−0.218 (0)	−0.217 (0)	−0.203 (0)	−0.132 (0.014)	−0.022 (0.683)	−0.043 (0.435)	−0.114 (0.042)	
Fees mean	−0.238 (0)	−0.238 (0)	−0.232 (0)	−0.206 (0)	−0.093 (0.085)	0.001 (0.983)	0.122 (0.03)	
Exchange outflow volume	−0.254 (0)	−0.25 (0)	−0.243 (0)	−0.153 (0.004)	−0.043 (0.432)	−0.047 (0.386)	−0.115 (0.04)	
Hash rate	−0.304 (0)	−0.31 (0)	−0.381 (0)	−0.443 (0)	−0.53 (0)	−0.595 (0)	−0.663 (0)	
In-house exchange volume	−0.309 (0)	−0.309 (0)	−0.327 (0)	−0.312 (0)	−0.189 (0)	−0.193 (0)	−0.225 (0)	
Over 1000	−0.341 (0)	−0.347 (0)	−0.372 (0)	−0.399 (0)	−0.439 (0)	−0.509 (0)	−0.634 (0)	
Difficulty	−0.391 (0)	−0.401 (0)	−0.454 (0)	−0.524 (0)	−0.599 (0)	−0.664 (0)	−0.713 (0)	
Purpose bitcoin ETF holdings	−0.428 (0)	−0.447 (0)	−0.558 (0)	−0.697 (0)	−0.741 (0)	−0.779 (0)	−0.81 (0)	
Thermocap	−0.443 (0)	−0.451 (0)	−0.5 (0)	−0.563 (0)	−0.637 (0)	−0.704 (0)	−0.767 (0)	
Block height	−0.477 (0)	−0.484 (0)	−0.524 (0)	−0.579 (0)	−0.648 (0)	−0.709 (0)	−0.767 (0)	
Over 10000	−0.516 (0)	−0.527 (0)	−0.58 (0)	−0.623 (0)	−0.676 (0)	−0.706 (0)	−0.687 (0)	
Supply last active 3+ years ago	−0.533 (0)	−0.541 (0)	−0.584 (0)	−0.624 (0)	−0.668 (0)	−0.708 (0)	−0.718 (0)	
UTXO Total	−0.574 (0)	−0.582 (0)	−0.629 (0)	−0.686 (0)	−0.747 (0)	−0.8 (0)	−0.842 (0)	
Addresses with non-zero balance	−0.599 (0)	−0.607 (0)	−0.652 (0)	−0.711 (0)	−0.77 (0)	−0.815 (0)	−0.835 (0)	
Over 0.1	−0.601 (0)	−0.604 (0)	−0.624 (0)	−0.656 (0)	−0.705 (0)	−0.752 (0)	−0.789 (0)	
Over 1	−0.603 (0)	−0.604 (0)	−0.612 (0)	−0.628 (0)	−0.666 (0)	−0.71 (0)	−0.744 (0)	
Over 0.01	−0.624 (0)	−0.629 (0)	−0.657 (0)	−0.699 (0)	−0.75 (0)	−0.796 (0)	−0.833 (0)	

From Fig. 8, it can be seen the previously identified high correlation with price, exhibiting higher volatility. However, in more recent months, when Bitcoin price has fallen considerably, NUPL seems to exhibit higher volatility. This could point to a smaller NUPL volatility and higher price volatility in bull markets and a reverse in bear markets, where NUPL volatility increases, and the price tends to stabilize. Although lower, the Spent Output Profit Ratio (SOPR) seems to exhibit some level of correlation with the price. It must be mentioned that both NUPL and SOPR show positive correlation coefficients, which point to a direct relationship between them and the price. Again, to better understand the relationship between SOPR and Bitcoin price, the time series are plotted in Fig. 9.

Figure 8 Normalized price and NUPL time series plot (X axis-days, Y axis-normalized value).

Figure 9 Normalized price and SOPR time series plot (X axis-days, Y axis-normalized value).

Unlike NUPL, SOPR seems to exhibit higher volatility in a bull market and tends to stabilize in a bear market. The correlation between SOPR and price is noticeable, but it is obvious that it isn’t a strong one, as seen in the case of NUPL. However, further study of the relationship between SOPR and price could unlock a greater understanding of how the two are connected and how SOPR could be leveraged to predict the price evolution of Bitcoin. Lastly, while negligent, the relationship between Twitter Sentiment and Bitcoin price will be studied. The two normalized time series are plotted in Fig. 10.

Figure 10 Normalized price and Twitter sentiment time series plot (X axis-days, Y axis-normalized value).

As stated above, no apparent relationship exists between the two. However, what is interesting is that Twitter sentiment seems to spike right before a price drop. This could point to many things, from overly optimistic holders that hope their investment will net a good return to mischievous players who seek to manipulate the sentiment and determine others to buy in so they can then dump their coins. It could also point to a mixture of the two scenarios or even other cases. Future studies could look deeper into the connection between social media and cryptocurrencies, by implementing more advanced NLP techniques.

The input data selection step follows this analysis phase. Different thresholds are set for the experiments, adding more data as the experiments are running.

Experimental setup

The experiments are run on an Intel I7 3,930 k CPU and an Nvidia GTX 1070 8 GB GPU machine. The Python version used was 3.9.12 and Tensorflow version 2.6.0. The proposed algorithm for the best variable combination identification is presented in Fig. 11.

Figure 11 Pseudo code for best parameter selection.

The algorithm works as follows. To select data, 12 thresholds are set from more to less restrictive. Six time-steps are used for each method, ranging from 1 to 14 days. The moving average windows range from 1 to 30. A total of 135 unique hyperparameter combinations are used to train the LSTM models. In total, 291,600 models are trained and benchmarked using the Adam optimizer and mean squared error (MSE) loss function over 100 epochs.

The proposed method aims to identify the best combination of metrics, including the correlation threshold, data transformation using the moving average sample, and deep learning parameters such as layer size, dropout, and learning rate. These combinations generate the best results, the lowest predictive error, for each selected time step. Table 4 presents all experimental parameters, including the 291,600 unique combinations resulting from these parameters. Coefficients for correlation threshold and dropout are shown, while the time step represents the number of days predicted, the moving average window represents the number of days used to compute the average, and the LSTM layer represents the dimensionality of the output space.

Table 4 Experimental hyperparameters values.

Parameter	Values	
Time step	[1, 3, 5, 7, 10, 14]	
Moving average window	[1, 3, 7, 14, 30]	
Correlation threshold	[0.95, 0.9, 0.85, 0.8, 0.75, 0.7, 0.65, 0.6, 0.55, 0.5, 0.45, 0.4]	
LSTM layer	[32, 64, 128, 256, 512, 1,024, 2,000, 2,500, 3,000]	
Dropout	[0.1, 0.2, 0.4, 0.6, 0.8]	
Learning rate	[0.01, 0.001, 0.0001]	

Results

The results of the trained models for predicting Bitcoin price are presented in Table 5, comparing the performance of the Proposed Approach (PA) and the Martingale (Mart) Approach Benchmark. The results reveal that the PA outperforms the Martingale approach in 3 out of the 6-time steps, with an average improvement of 9.3% in Mean Absolute Percentage Error (MAPE). Table 6 shows the selected metrics for each threshold, which demonstrates the best combination of variables for each time step.

Table 5 The proposed approach (PA) vs martingale (Mart) approach benchmark. (BOLD represents the best results for each Time step.)

Time step	PA
RMSE	PA
MAE	PA MAPE	Mart
RMSE	Mart
MAE	Mart
MAPE	PA parameters	
1	1,442.34	1,056.34	0.03	1,533.00	1,191.99	0.04	1&0.7&1k&0.6&0.01	
3	3,330.30	2,682.92	0.08	2,453.35	1,828.75	0.06	1&0.8&1k&0.4&0.01	
5	5,243.44	4,413.47	0.14	4,876.21	4,057.77	0.13	1&0.9&3k&0.6&0.001	
7	6,568.82	5,666.63	0.17	6,200.79	5,333.31	0.16	1&0.9&3k&0.6&0.001	
10	7,973.67	7,105.18	0.21	8,357.94	7,520.11	0.21	7&0.6&3k&0.4&0.01	
14	6,058.08	5,025.92	0.15	10,422.04	9,648.01	0.26	14&0.4&1k&0.2&0.001	

Table 6 Selected metrics for each threshold.

Threshold	Metrics	
0.9	Market Cap to Thermocap Ratio, MVRV, Market Cap, Price	
0.8	Market Cap to Thermocap Ratio, MVRV, NUPL, Market Cap, Price	
0.7	Market Cap to Thermocap Ratio, MVRV, NUPL, Market Cap, Price	
0.6	Over 0.01, Over 0.1, Over 1, Inter-Exchange Transfers, Market Cap to Thermocap Ratio, Suply Last Active 1+ Years Ago, MVRV, NUPL, Market Cap, Price	
0.4	Addresses with Non-zero Balance, Over 0.01, Over 0.1, Over 1, Over 100, Over 10,000, Inter-Exchange Transfers, Thermocap, Market Cap to Thermocap Ratio, Purpouse Bitcoin ETF Holdings, Bitcoin Fund Holdings, 3iq Holdings, Suply Last Active 1+ Years Ago, Suply Last Active 3+ Years Ago, MVRV, SOPR, NUPL, Block Height, UTXO Total, Market Cap, Price	

For the 1-day time step, the best-performing model includes Price, Market Cap, MVRV, NUPL, and Market Cap to Thermocap Ratio variables. The best-performing models for the 5- and 7-days time-steps require more restrictive thresholds, retaining only highly correlated variables as inputs. As the time step increases, the threshold for improvement decreases, with a maximum of 40%. Figures 12–14 further illustrate the performance of the best-performing models in predicting Bitcoin price evolution.

Figure 12 Best performing models for 1 day (A) & 3 days (B) (X axis–days, Y axis–bitcoin price).

Figure 13 Best performing models for 5 days (A) & 7 days (B) (X axis–days, Y axis–bitcoin price).

Figure 14 Best performing models for 10 days (A) & 14 days (B) (X axis–days, Y axis–bitcoin price).

When comparing the results of the PA with Chen (2023), the PA achieves superior results in both root mean squared error (RMSE) and MAPE, as shown in Table 7. The improvement in performance is attributed to the enhanced data collection, data analysis and processing, and model selection and hyperparameter tuning steps.

Table 7 The proposed approach (PA) vs Chen (2023) approach benchmark. (BOLD representes the best result for each metric-RMSE and MAPE.)

Model	RMSE	MAPE	
PA	1,442.34	0.03	
Chen–Period 1	321.61	0.0339	
Chen–Period 2	2,096.24	0.0329	

Discussion

Comparison with the martingale approach

In the field of financial market research, the Martingale model serves as a foundational benchmark for assessing market efficiency and predictability (Smith, 2009; Richard & Vecer, 2021). The Martingale model postulates that future price movements are entirely independent of past prices, implying that no investment strategy can yield returns exceeding those achieved by random chance. In this study, the Martingale model is used as a baseline for comparison to evaluate the effectiveness of the PA, which utilizes on-chain metrics and Twitter sentiment data. If the proposed models outperform the Martingale model, it suggests that the analyzed variables contain predictive information about Bitcoin’s price movements, thus offering the potential for traders to achieve better returns. By contrasting the findings against this well-established benchmark, the aim is to provide a rigorous assessment of the unique predictive capabilities introduced by the selected variables.

The proposed approach outperforms the Martingale method in predicting Bitcoin prices, yielding a lower average MAPE score for the 6-time steps. The most significant improvement is observed in longer time steps, such as a 60% enhancement for the 14-day time step. The Martingale model performs well for predicting the next day and three-day prices, but its performance deteriorates as the lag between the present price and the predicted price increases. On average, the proposed approach shows a 9.3% improvement in MAPE compared to the Martingale method. Therefore, RQ4 is answered. By implementing a combination of on-chain metrics and Twitter sentiment, an increase of 9.3% in prediction performance is achieved.

Thresholds and model performance

For the 5- and 7-day models, the best performance is achieved when the threshold is more restrictive, retaining only the highly correlated variables in the input. As the prediction horizon increases and the predictive quality of the present price decreases, using a broader range of metrics improves the model’s performance. Consequently, less restrictive thresholds can lead to improved results when working with longer time steps.

For the 1-day time step, the best performance is obtained when the model is trained with Price, Market Cap, MVRV, NUPL, and Market Cap to Thermocap Ratio variables. While the present-day Price offers sufficient information for high-quality predictions, other metrics enhance performance albeit not substantially.

For the 10-day time step, additional metrics are incorporated into the predictive model, including Over 0.01, Over 0.1, Over 1, Inter-Exchange Transfers, and Supply Last Active 1+ Years Ago. The inclusion of Supply Last Active 1+ Years Ago, an important metric for the Bitcoin bear market, improves predictive performance. As the time step increases, the threshold for improvement decreases, with a maximum of 40%.

Long-term predictions

The proposed approach yields reliable results for long-term predictions, such as 10 or 14 days ahead, albeit less accurate. These outcomes encourage further research, showcasing the potential performance when on-chain metrics are input into a deep learning model.

Comparison with Chen (2023)

The best comparison is achieved when benchmarking with Chen (2023)–Period 2, due to the varying time frames used in the analysis of the two papers. When comparing the two approaches, the PA achieves superior results in both RMSE and MAPE, with improved performance in a smaller data sample by a factor of 4. The performance improvement can be attributed to several factors: enhanced data collection, better data analysis and processing using a more advanced sentiment extraction technique (logistic regression vs VADER), and the model selection and hyperparameter tuning process that ensured optimal parameter selection. This last factor is particularly important, as the performance of LSTM models is often highly sensitive to hyperparameter choices, making the tuning process both challenging and time-consuming. By presenting a better Bitcoin price prediction method, the practical implications of this study are that investors and other stakeholders can make more informed decisions.

Comparison with literature

When comparing to the related literature, this work adds a new layer by introducing a wide range of on-chain metrics alongside Twitter sentiment, providing a more holistic model. Furthermore, prior works focused largely on investor sentiment and attention as isolated factors (Guegan & Renault, 2021; Li et al., 2022). This study synthesizes these with on-chain metrics, creating a more comprehensive analysis framework. In reference to time scale, previous articles largely indicated that their models or factors were mainly significant in the short term (Bouteska, Mefteh-Wali & Dang, 2022). The present research offers insights into more long-term behavior, especially during the transition from bull to bear markets.

As such, the novelty of this study lies in its multi-dimensional approach to predicting Bitcoin prices. While previous research has separately examined on-chain metrics or investor sentiment, ours is the first to combine these in a complementary manner. Compared to work of Jay et al. (2020), our study expands the data extracted from Twitter, by introducing the sentiment component. As such, this is the first study, to our knowledge, to combine on-chain metrics with social media sentiment and it further bridges the gap between social media sentiment and the crypto market. We believe this is an evolution in the field, as it culminates the work done up to this point and lays new paths going further. This enables a more comprehensive understanding and prediction model that captures both technical and psychological market factors, filling a gap in existing literature. Therefore, future works can build upon the research presented in this article and further analyze the interconnectivity between social media data and blockchain data, to further improve the performance of prediction models. For industry stakeholders, the work presented in this article could facilitate the implementation of the PA to yield improved returns and to gain a better understanding of the forces that drive the cryptocurrency market. Therefore, if financial gain is to be had from the PA, additional ethical concerns should be accounted for.

Novelty and impact on field of study

This research presents a new methodology for forecasting Bitcoin prices by combining on-chain metrics with Twitter sentiment data, resulting in a distinctive combination of transactional and public sentiment indicators. Our research enhances the predictive accuracy and captures the dynamic nature of cryptocurrency markets by contrasting with traditional models such as the Martingale. The results have important implications for both scholarly discussions and real-world implementations, offering investors, stakeholders, and researchers a comprehensive instrument that encompasses the diverse factors influencing cryptocurrency prices in the digital era.

As such, our research distinguishes itself within the field of cryptocurrency studies by presenting a unique approach that combines on-chain metrics and Twitter sentiment data to forecast the price of Bitcoin. In the past, scholarly investigations in this field have frequently focused on analyzing these data sources individually. In the context of blockchain analysis, on-chain metrics provide valuable information regarding transaction volumes, active addresses, and other activities specific to the blockchain. However, it is important to acknowledge that these metrics may overlook the emotional and psychological aspects that can exert an impact on price dynamics. In a synergistic manner, the analysis of sentiment in Twitter data offers valuable insights into the overall mood and perceptions of the wider community. However, it is important to note that this analysis lacks the empirical support provided by tangible transactional data. By integrating these two components, our model effectively encompasses both the quantitative factors influencing Bitcoin’s fluctuations and the qualitative sentiments that underlie its perceived worth.

Limitations and future work

A potential weakness of the study lies in its reliance on deep learning models, which tend to shift data and predict future prices as the present value. While the PA addresses this issue and generates improved results, predicting the price for large time gaps remains highly challenging. This limitation necessitates further research to optimize the performance of deep learning models when they utilize on-chain metrics, and to explore alternative modeling techniques that might complement the current approach.

Some possible directions for future work include: Investigating the use of ensemble methods, which combine the predictions of multiple deep learning models, to improve the overall prediction accuracy and robustness. This strategy could potentially mitigate the shortcomings of relying on a single deep learning model for predicting prices over large time gaps.

Exploring the use of other statistical and machine learning techniques, such as support vector machines, decision trees, Bayesian models, or Hidden Markov Models to complement deep learning models. These alternative methods might provide different perspectives on the data, and help identifying different market regimes or trends, such as bullish, bearish, or sideways movements.

Incorporating additional external factors, such as macroeconomic indicators or news sentiment, to better capture the complex dynamics of the cryptocurrency market. This could help improve the predictive power of the model, especially for longer time horizons.

Expanding the selected time interval, which currently serves as a snapshot of Bitcoin’s history. The selection process for on-chain metrics could also be broadened. Insufficient data can result in overfitting or poor generalization.

Experimenting with different architectures or configurations of deep learning models, such as attention mechanisms, to better capture the temporal relationships between on-chain metrics and future price movements. This could potentially address the issue of models predicting future prices as the present value.

Conducting comprehensive evaluations of the model’s performance across different market conditions, such as bull and bear markets, could offer insights into the robustness of the proposed approach and identify areas for improvement.

Besides these broader limitations, there are some specific to this study. One such limitation is the use of a single platform, Twitter, for sentiment analysis. As a result, the sentiments captured may not fully represent the broader investor sentiment that could be gathered from multiple social media platforms. Another limitation is the choice of on-chain metrics, which, while expansive, is not exhaustive and may overlook other potential indicators of Bitcoin price. Another limitation is the sole use of the Pearson correlation to analyze the relationship between data and to establish the information cutoff. The results might be sensitive to the threshold cutoffs that were put in place. Additionally, the Pearson correlation measures only linear relationships and does not capture non-linear dependencies that could exist between the variables. To address this, future studies could incorporate additional measures of dependence, such as the Spearman rank correlation for monotonic relationships and potentially mutual information for more general dependencies. This will provide a more comprehensive view of the relationships between the variables.

By tackling the identified shortcomings and exploring the proposed directions for future studies, we could enhance the predictive capabilities of deep learning models in forecasting cryptocurrency prices over longer time spans.

Ethical considerations

Throughout the course of this study, various ethical dimensions were thoroughly examined to uphold the integrity and dependability of the results. One of the main focal points revolved around the safeguarding of data privacy and the maintenance of user anonymity. Due to the inherent characteristics of the data obtained from Twitter, precautions were undertaken to guarantee the anonymization of identifiable information. Moreover, the utilization of the data was conducted in strict adherence to the guidelines specified by the platform. The maintenance of data collection transparency was achieved through the provision of explicit information regarding the sources and methodologies employed, thereby ensuring the ability to reproduce and comprehend the process. Furthermore, the economic and societal ramifications of our predictive models were deliberated, considering the potential tangible effects on market dynamics and wider economic environments.

Due to the significant implications associated with financial predictions, the research was conducted with a rigorous commitment to accuracy in reporting, aiming to avoid any potential misrepresentation of information for stakeholders. While the primary objective of the study was impartial, we were mindful of the possible misapplication of financial forecasting instruments, such as in the context of manipulative market strategies. To achieve a harmonious equilibrium between safeguarding data privacy and promoting scientific transparency, methodologies and code were shared to facilitate the replication of research while upholding the integrity of the data.

Potential for replication

The study presents its methodologies and findings in a manner that aims to facilitate and promote future replication endeavors. The importance of replication in validating and strengthening research findings, especially in rapidly evolving fields like cryptocurrency analysis, is widely recognized and encouraged. Through the replication of this study across various contexts and conditions, the scientific community can assess the strength and reliability of the presented methodology, investigate potential subtleties, and make additional contributions to the ongoing discourse. We strongly support and promote such initiatives, as we believe they serve to validate our research and drive progress in the field by promoting a collaborative and iterative research approach.

Conclusions

This article offers two main theoretical contributions: Firstly, it enhances the understanding of the factors that influence Bitcoin’s price movements. Secondly, it puts forward a more effective approach to forecasting Bitcoin’s price. By scrutinizing various on-chain metrics in diverse market circumstances, this study has revealed potential market indicators such as turning points or alerts for a 5% price variation. Compared to the Martingale baseline and previous research, the suggested approach has produced better results. This improvement is due to the expanded range of analysis metrics, the refinement of the processing techniques, and the model development methodology proposed in this study.

The managerial implications that derive from this work enable different stakeholders, such as investors or legislators, to extract more and better information from existing data and make better decisions. Stakeholders involved in the decision-making process should follow the work presented in this article, the findings, and the proposed methodology to better understand the factors that influence the evolution of Bitcoin and respond accordingly.

Our study has some drawbacks, such as a restricted timeframe, utilization of a sentiment extraction method that enhances prior implementations but might not represent the most advanced approach for sentiment analysis, and incomplete on-chain metric selection. Future research can expand the time interval, enhance sentiment extraction using advanced NLP models, integrate more metrics, and improve forecasting models for better Bitcoin price predictions.

In closing, the contribution of this research should be reiterated. This is the first study to incorporate many on-chain metrics for Bitcoin and Twitter sentiment extracted with a superior method compared to previous implementations. These improvements have shown to yield great results and have proven that highly accurate Bitcoin price prediction is achievable. Furthermore, new insights have been extracted regarding the behaviour of the Bitcoin market, especially during the transition period from a bull to a bear market. These findings enable a better understanding of the market and an improved ability to predict its behaviour.

Additional Information and Declarations

Competing Interests

Author Contributions

Data Availability

The authors declare that they have no competing interests.

Alexandru Costin Baroiu conceived and designed the experiments, performed the experiments, analyzed the data, performed the computation work, prepared figures and/or tables, authored or reviewed drafts of the article, and approved the final draft.

Vlad Diaconita conceived and designed the experiments, analyzed the data, prepared figures and/or tables, authored or reviewed drafts of the article, and approved the final draft.

Simona Vasilica Oprea analyzed the data, authored or reviewed drafts of the article, and approved the final draft.

The following information was supplied regarding data availability:

The data and code is available at Zenodo: Alexandru Costin Baroiu, Vlad Diaconita, & Simona Vasilica Oprea. (2023). Bitcoin volatility in bull vs. bear market–insights from analyzing on-chain metrics and Twitter posts (Version 1) [Data set]. Zenodo. https://doi.org/10.5281/zenodo.7791503.

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
