# Peer review of "Bitcoin volatility in bull vs. bear market-insights from analyzing on-chain metrics and Twitter posts"

_PeerJ Computer Science, doi:10.7717/peerj-cs.1750_

## Round 0.1 · original submission · Major Revisions

Dear Dr. Diaconita,

Thank you for your submission to PeerJ Computer Science.

It is my opinion as the Academic Editor for your article - Bitcoin volatility in bull vs. bear market - insights from analyzing on-chain metrics and Twitter posts - that it requires a number of Major Revisions

**Language Note:** PeerJ staff have identified that the English language needs to be improved. When you prepare your next revision, please either (i) have a colleague who is proficient in English and familiar with the subject matter review your manuscript, or (ii) contact a professional editing service to review your manuscript. PeerJ can provide language editing services - you can contact us at copyediting@peerj.com for pricing (be sure to provide your manuscript number and title). – PeerJ Staff

·

Basic reporting

The topic of the paper is interesting and up-to-date.
I have following remarks connected with the text:
• I think the English level is ok.
• The literature basis is not enough especially it lacks the newest paper from last 2-3 years from good international journals.
• The article structure is ok.
• The results are connected with goals of the paper.

Experimental design

• The research is within aim and scope of the Journal.
• Research question are ok, but it could be possible to add some more research questions.
• Methods are describe in good way and are understandable foe a reader.

Validity of the findings

• The discussion is short and without in-deep analysis of relations results with international papers – you should more discus results with them.
• The limitation is very puzzled and generals – the Authors should add some limitation specific for the paper.
• Please describe more about novelty of the findings in the paper.

Reviewer 2 ·

Basic reporting

The paper has interesting findings, the topic is relevant and present a lot of technical work developed
My suggestions for the paper are
1) the selected period of time seems arbitrary, tweets are selec from july 1, 2021 until June 30 2022 (line 254). Could the authors explain more about the selection of the period?
2) Along the paper, the discussion is based on the correlation for the selection of the metrics. In line 275 is mentioned that only metrics with correlation greter than 0.8 are presented. Please explain more about this selection ( why 0.8?). Additionally, Pearson correlation is a linear nmeasure of dependence. Please , consider comments about the existence of non-linear dependence, which is not reflected in Pearson correlation
3) In my opnion is necessary a paragraph explainig the martingale model and how you are applying in this context
4) In lines 255 and 256 you selected tweets with the restriction of not have been retweetd. Please explain the reason

Experimental design

The selection of metrics, the restrictions for the tweets selected in my opinion are in general correct. Please attend my observations in the past section

Validity of the findings

The findings are very interesting and solid with the techniques appled.
The are restrictions in the sense that it is not possible to know what will happen in the case of a comparative study with different algorithms ( like vector machines an so on ). This fact is recognized by the authors in the section of limitations of the work ( this section is very well structured)

Additional comments

Please attend the suggestinos in the section "Basic Reporting"

---

## Round 0.2 · Major Revisions

Please adjust the manuscript accordance with reviewer comments.

·

Basic reporting

Clear and unambiguous, professional English used throughout: The discussion section is written in clear, professional English. It presents the study's findings and their implications effectively.

Literature references, sufficient field background/context provided: The discussion references prior literature, especially when comparing the research to the Martingale model and other studies. It provides context for the findings.

Professional article structure, figures, tables: The discussion section follows a professional structure by discussing the study's results, comparing them with other models, and suggesting future directions. It does not require figures or tables in this context.

Self-contained with relevant results to hypotheses: The discussion is self-contained and presents relevant results, comparisons, and future directions in line with the study's objectives.

Experimental design

Original Primary Research within Aims and Scope of the Journal:

The text appears to describe original research that falls within the aims and scope of the journal. It discusses the use of on-chain metrics and Twitter sentiment data for predicting Bitcoin prices, which is a relevant and timely topic in the field of cryptocurrency research.

Research Question Well Defined, Relevant & Meaningful:

The research question is reasonably well defined. It focuses on assessing the effectiveness of a proposed approach (PA) for predicting Bitcoin prices using on-chain metrics and Twitter sentiment data. The relevance and meaningfulness of the research question are evident as it addresses the practical application of cryptocurrency price prediction, which is of interest to investors and stakeholders.

Identification of Knowledge Gap and Contribution:

The text mentions the use of the Martingale model as a benchmark and implies that the research aims to demonstrate whether the proposed approach outperforms this model. This suggests an identification of a knowledge gap related to the effectiveness of different prediction models for Bitcoin prices.
However, the text could be more explicit in stating how the study contributes to filling this knowledge gap. It mentions improved results but does not elaborate on the specific contributions or implications of the research beyond stating that it yields better results. A clearer statement of the research's contribution to the field would enhance this aspect.

Rigorous Investigation and Ethical Standards:

The text does not provide detailed information about the specific methods used in the research, making it challenging to assess the rigor of the investigation or whether ethical standards were adhered to. For a comprehensive evaluation, the full research paper would need to provide in-depth information on data collection, analysis, and ethical considerations.

Methods Described with Sufficient Detail to Replicate:

The text does not provide sufficient detail about the methods used in the research. It mentions the utilization of on-chain metrics and Twitter sentiment data but lacks specifics regarding data sources, data preprocessing, modeling techniques, and evaluation methods. To ensure replicability, a full research paper would need to include detailed descriptions of these aspects.

Validity of the findings

Impact and Novelty Not Assessed:

The text does not assess the impact or novelty of the research. It primarily focuses on describing the methodology and findings related to predicting Bitcoin prices using on-chain metrics and Twitter sentiment data. There is no explicit discussion of how the research advances the field or addresses a gap in the literature.

Encouragement of Meaningful Replication:

The text does not explicitly mention replication studies or their rationale. It primarily presents the research's approach, findings, and limitations but does not discuss the potential for replication or how replication could benefit the literature. Encouraging replication and explaining its value to the field would enhance this aspect.

Underlying Data Availability:

The text does not mention the availability of underlying data or whether the data used in the research is stored in a discipline-specific repository. For transparency and to allow other researchers to assess the robustness of the findings, it is essential to provide information about data availability and repository location.

Conclusions Well Stated and Linked to Research Question:

The text provides conclusions related to the research findings, particularly regarding the comparison with the Martingale model and the performance of the proposed approach. The conclusions appear to be linked to the research question, which focuses on the effectiveness of the prediction model.

Reviewer 2 ·

Basic reporting

I reviewed the changed applied to the paper. In my opinion the paper has achieved a high quality to be published in Peer J. Computer Science

Experimental design

The research questions are well defined and the methodology is very rigorous. My suggestiones were applied and the paper should be accepted for publication

Validity of the findings

I reviewed the changed applied to the paper. In my opinion the paper has achieved a high quality to be published in Peer J. Computer Science.
The research questions are well defined and the methodology is very rigorous. My suggestiones were applied and the paper should be accepted for publication

Additional comments

No additional comments

---

## Round 0.3 · Major Revisions

Dear Authors,

Please revise your manuscript to address the reviewer's suggestions.

·

Basic reporting

The provided text appears to use professional and technically correct language.
The structure seems to adhere to professional standards
The paper is self-contained and present relevant results to hypotheses.

Experimental design

The knowledge gap is not explicitly stated. However, it refers to investigating the relationship between on-chain metrics and Twitter sentiment, potentially contributing to the understanding of Bitcoin price movements.
Content touches on aspects related to the research question and potential knowledge gap, but a more thorough assessment would require a review of the complete paper. The availability of detailed methods and information about the ethical standards followed is essential for a comprehensive evaluation.

Validity of the findings

Paper lacks clear section headings and a well-defined structure, making it challenging to follow the flow of the paper. A clear and organized structure is crucial for effective communication.
While the text mentions the correlation between various metrics and Bitcoin prices, the absence of figures or specific data points makes it difficult to assess the strength of these correlations. Including relevant figures and data is crucial for supporting research findings.
The text provides results but does not offer detailed interpretation or discussion. A thorough analysis of the results, along with their implications, is necessary for a comprehensive understanding of the research outcomes.
The language used in the textcould be more concise and formal. Professional writing standards, including adherence to grammatical and stylistic conventions, contribute to the credibility of the paper.

---

## Round 0.4 · accepted · Accept

I am writing to inform you that your manuscript - Bitcoin volatility in bull vs. bear market - insights from analyzing on-chain metrics and Twitter posts - has been Accepted for publication. Congratulations!